# Small Molecule HIV-1 Attachment Inhibitors: Discovery, Mode of Action and Structural Basis of Inhibition

**DOI:** 10.3390/v13050843

**Published:** 2021-05-06

**Authors:** Yen-Ting Lai

**Affiliations:** Vaccine Research Center, National Institute of Allergy and Infectious Diseases, National Institutes of Health, Bethesda, MD 20892, USA; yen-ting.lai@nih.gov

**Keywords:** fostemsavir, temsavir, attachment inhibitors, HIV-1 entry, crystal structures, antiviral activity

## Abstract

Viral entry into host cells is a critical step in the viral life cycle. HIV-1 entry is mediated by the sole surface envelope glycoprotein Env and is initiated by the interaction between Env and the host receptor CD4. This interaction, referred to as the attachment step, has long been considered an attractive target for inhibitor discovery and development. Fostemsavir, recently approved by the FDA, represents the first-in-class drug in the attachment inhibitor class. This review focuses on the discovery of temsavir (the active compound of fostemsavir) and analogs, mechanistic studies that elucidated the mode of action, and structural studies that revealed atomic details of the interaction between HIV-1 Env and attachment inhibitors. Challenges associated with emerging resistance mutations to the attachment inhibitors and the development of next-generation attachment inhibitors are also highlighted.

## 1. Introduction

Human immunodeficiency virus type 1 (HIV-1), a retrovirus that integrates its genetic information into host cells upon infection, can lead to acquired immunodeficiency syndrome (AIDS) if not treated [1]. Currently, there are roughly 38 million people globally living with HIV-1 (UNAIDS 2020 Fact Sheet) [2]. Seven classes of antiretroviral drugs, including protease inhibitors [3,4], nucleoside/nucleotide reverse transcriptase inhibitors (NRTIs) [5], non-nucleoside reverse transcriptase inhibitors (NNRTIs) [6], integrase inhibitors [7,8], post-attachment inhibitors [9], CCR5 antagonists [10], and fusion inhibitors [11] have been currently approved for the treatment of HIV-1 infection. Due to the high genetic diversity of HIV-1, monotherapy of any of the approved HIV-1 treatments usually leads to a selection of resistance mutations [12]. Combination antiretroviral therapy (cART) that utilizes drugs from more than one class of HIV-1 drugs has proven to be very effective in controlling viral loads in HIV-1^+^ patients, rendering HIV-1 infection a lifelong chronic disease that is manageable for patients who have access to cART (~26 million patients based on the UNAIDS 2020 Fact Sheet) [13,14]. However, after decade-long cART treatment, patients can develop resistance to multiple classes of currently approved drugs and face viral rebound and disease progression [13,14].

A newly FDA-approved antiretroviral drug, fostemsavir (brand name Rukobia), represents a new class of inhibitors referred to as attachment inhibitors. In the literature, the term “entry inhibitor” has sometimes been used to collectively refer to the four classes of inhibitors that target the HIV-1 entry process, including attachment inhibitors, post-attachment inhibitors, CCR5 antagonists, and fusion inhibitors (Table 1). Fostemsavir and other attachment inhibitors block HIV-1 entry by using a unique mechanism leading to low cross-resistance with other antiretroviral drug classes and thus was approved for use in highly treatment-experienced patients who develop drug resistance to currently approved drugs and have limited treatment options. The discovery, development, mode of inhibition, and drug resistance of fostemsavir and related attachment inhibitors are reviewed in this article.

## 2. HIV-1 Entry into Host Cells

The HIV envelope protein (Env) is the only glycoprotein displayed on the surface of the HIV virion [15,16,17]. Env forms a trimer, where each protomer is composed of a heterodimer of gp120 and gp41 subunits that non-covalently associate together [18,19]. The gp120 subunit is responsible for recognizing and binding to the receptor CD4 on CD4^+^ T cells and macrophages [20]. The binding of CD4 leads to conformational changes and exposure of the coreceptor binding site, which can then engage the coreceptors, CCR5 or CXCR4 [21,22]. The binding of CD4 and a coreceptor result in the shedding of gp120, and subsequent conformational changes in gp41 lead to the fusion of viral and host cell membrane, allowing the entry of the HIV capsid and genome into host cells [23,24]. The entry process (Figure 1) is a critical aspect of the HIV-1 life cycle and a target of many therapeutic strategies, including but not limited to small molecule inhibitors and antibody modalities [25,26].

### FDA-Approved Drugs Targeting HIV-1 Entry

Two antiretroviral drugs, maraviroc and enfuvirtide, target different steps of the HIV entry process. Maraviroc, which is a CCR5 antagonist, binds to the coreceptor CCR5 and prevents CCR5 from binding to the CD4-bound Env [27]; enfuvirtide binds to gp41 and prevents the conformational changes of gp41 that are required for membrane fusion to occur [28]. Significant limitations are associated with the use of maraviroc and enfuvirtide—maraviroc is only effective against CCR5-tropic HIV-1 viruses, and enfuvirtide requires twice-daily intravenous injection due to its nature as a peptide drug.

An antibody inhibitor, ibalizumab, was approved by the US FDA in 2018 for the treatment of HIV-1 infection in patients who developed multi-drug resistance. Ibalizumab binds to domain 2 of the CD4 receptor and thus is classified as a post-attachment inhibitor which allows HIV-1 Env to bind to domain 1 of CD4 but prevents the downstream steps required for the entry to occur [29]. Due to its unique mode of action compared to other entry inhibitors, ibalizumab has been designated a first-in-class medication by the US FDA.

## 3. Discovery, Clinical Trials, and FDA Approval of Fostemsavir

Fostemsavir was approved by the FDA in 2020 for the treatment of HIV-1 infection in heavily treatment-experienced patients who developed resistance to multiple existing antiretroviral drugs. Temsavir (previously BMS-626529), the active compound of the prodrug fostemsavir (previously BMS-663068/GSK-3684934), was discovered by a research group in Bristol Myers Squibb in 2003 [30,31]. A phenotypic inhibition assay was set up to screen libraries of compounds that can prevent a pseudotyped reporter virus, which has an HIV-1 LAI-∆env-luc backbone supplemented with a JRFL (an M-tropic HIV-1 strain) Env protein in trans, leading to virions that were only capable of one round of replication, from infecting HELA cells expressing a CD4 receptor and a CCR5 coreceptor. An indole glyoxamide compound was identified to have strong inhibition of luciferase expression in this phenotypic assay. 

Due to the nature of the phenotypic inhibition assay, the identified inhibitor could inhibit the pseudotyped virus at various stages, including the entry and capsid uncoating processes, the reverse transcription of viral mRNA to complementary DNA, integration of the complementary DNA into the host genome, and the transcription and translation of the luciferase reporter protein. In addition, because the target HELA cells expressed CCR5 coreceptor, the inhibitors identified could specifically target the interactions between CCR5 and HIV-1 Env and thus be dependent on the tropism of specific HIV-1 viruses. Extensive studies were carried out to elucidate the mode of action of the indole glyoxamide. Several lines of evidence suggested that the indole glyoxamide inhibits the HIV-1 envelope protein: (1) the compound inhibits infection of HELA cells expressing CD4 and CXCR4 by pseudotyped viruses derived by T-tropic HIV-1 LAI strain; (2) in biochemical assays, the compound showed no inhibition to HIV-1 reverse transcriptase, protease, and integrase; (3) the time-of-addition in a single-cycle infection assay showed that the indole glyoxamide had reduced activity if added >30 min after infection, suggesting the compound inhibits HIV-1 at an early stage of the life cycle; (4) the compound is capable of inhibiting a virus-free cell-based fusion assay where the fusion of cells expressing HIV-1 envelope protein and those expressing CD4 and CXCR4 is inhibited in the presence of the compound. It was concluded that the target of the indole glyoxamide is the HIV-1 envelope protein [30].

### Clinical Development of Indole Glyoxamide Inhibitors

The initially identified indole glyoxamide was systematically optimized by structure-activity relationship (SAR) studies, leading to a series of derivatives and analogs that had significantly improved antiviral activities and pharmaceutical profile [31]. Three compounds derived from the indole glyoxamide, including BMS-378806, BMS-488043, and BMS-626529, had entered clinical trials for safety and efficacy studies. Although only the three indole glyoxamide-derived compounds that entered clinical trials are reviewed here, it is noted that a tremendous amount of efforts were devoted to optimizing the medicinal chemistry of temsavir. A comprehensive review of the SAR studies that were carried out to obtain the potent temsavir from the initially identified indole glyoxamide can be found in [32]. 

The first clinical candidate derived from the indole glyoxamide, BMS-378806, has a 4-methoxy 7-azaindole linked to 2-(R)-methylpiperazine through the oxoacetyl functional group identified in the original indole glyoxamide (see Figure 2 for structures). BMS-378806 exhibits no tropism specificity and significantly improved potency across a panel of clinical isolates compared to the original indole glyoxamide in vitro; however, virus passage under the presence of a high concentration of BMS-378806 showed that the virus can establish resistance with mutations M426L, M434I/V, and M475I [33,34]. Furthermore, there is a 30-fold variability in potency across the HIV-1 strains tested [33]. Phase I dose-escalation studies in healthy adults showed generally good tolerability of BMS-378806, but the plasma concentration of the drug was below that targeted for efficacy studies [35], and thus further development of BMS-378806 was halted.

Further medicinal chemistry optimization led to the discovery of BMS-488043, a 4,7-dimethoxy-azaindole with a nitrogen at the C6 position of the azaindole and the remainder of the compound identical to the original indole glyoxamide (Figure 2). BMS-488043 exhibited a moderate improvement in antiviral potency and a significant improvement in pharmaceutical profile [36], including increased in vitro metabolic stability and membrane permeability and a better pharmacokinetic profile when compared to BMS-378806. In a Phase I clinical trial, it was determined that dosing BMS-488043 with a high-fat meal led to a linear dose-response of plasma drug concentration. Thus, in a following clinical trial testing the efficacy of BMS-488043 in HIV-1-infected patients, doses of 800 and 1800 mg of BMS-488043 with a high-fat meal following a twice-daily schedule were selected to provide proof of concept of an attachment inhibitor as monotherapy in an 8-day period [37]. On day 8, the mean decline of plasma viral load was 0.72 and 0.96 log_10_ copies/mL for the 800 mg and 1800 mg dose groups, respectively, representing a significant decrease when compared to the placebo group (a decrease of 0.02 log_10_ copies/mL). Thus, the attachment inhibitor was approved to be effective in reducing viral load as a monotherapy. However, the emergence of viral resistance was observed in four out of thirty subjects, evidenced by the more than 10-fold reduction of viral susceptibility by the end of the dosing period. Similar to what was observed in the in vitro passage experiment in the presence of BMS-378806 (see above), the M426L of HIV-1 Env was identified as one of the emerging resistance mutations in HIV-1 infected patients treated with BMS-488043. In addition, V68A, L116I, and S375I/N were also identified to correlate with resistance based on population-based sequencing of the HIV-1 Env gene [38]. 

Despite the efficacy of BMS-488043 in reducing viral loads in a monotherapy clinical trial, the needs to dose the drug with a high-fat meal to increase the solubility of BMS-488043 in plasma and the high dose level (800–1800 mg twice daily) required to achieve the target plasma concentration were significant challenges for further development. A solution to the solubility issue was found in the phosphonooxymethyl-based prodrug approach [36], which was previously utilized in antifungal drug candidates [39]. The prodrug has increased solubility in the gastrointestinal tract due to the polar phosphonooxymethyl functional group and was designed to be cleaved by an alkaline phosphatase present on the brush border membrane to release the parent drug before absorption by intestine endothelial cells. Indeed, the BMS-488043 prodrug showed a 300-fold improvement in solubility (12 mg/mL at pH = 5.4 versus 0.04 mg/mL at a pH range from 4 to 8) compared to the parent drug BMS-488043. A clinical pharmacokinetic study showed that the BMS-488043 prodrug led to the dose-proportional exposure of BMS-488043 in the plasma for doses ranging from 25 to 800 mg. At the equivalent dose of 800 mg, the BMS-488043 prodrug showed a 6-fold higher C_max_ (maximum concentration) in plasma than the parent drug dosed with a high-fat meal and a 3-fold higher AUC (area-under-curve). However, the t_max_ of the BMS-488043 prodrug is much earlier, between 0.5 and 1 h, with a half-life of 1.5 ± 0.2 h, much shorter than the parent drug dosed in a capsule formulation (~10 h), indicating that an extended-release formulation of the BMS-488043 prodrug will be beneficial for a longer protection window [36].

While BMS-488043 and its prodrug were tested in clinical trials, a more potent compound, BMS-626529 (temsavir), was discovered through thoughtful SAR campaigns [40]. BMS-626529 differs from BMS-448043 in that BMS-626529 has a 3-methyl-1,2,4 triazole linked to the C7 position of the C6 substituted azaindole, where the C7 position on the azaindole of BMS-448043 was linked to a methoxy functional group (Figure 2). It has been concluded from extensive SAR studies that derivatives possess higher antiviral potency when the functional group linked to the C7 position maintains a coplanar arrangement with the indole heterocycle [41]. BMS-626529 has an overall ~10-fold improvement in antiviral potency when tested in vitro against a panel of laboratory-adapted and clinical HIV-1 isolates. Notably, HIV-1 clade AE viruses are resistant to BMS-626529, as well as its predecessors, including BMS-488043 and BMS-378806 [42]. 

The significantly improved antiviral potency and well-behaved pharmaceutical profile of BMS-626529 rendered this compound a favorable clinical candidate over BMS-488043 [43]. The phosphonooxymethyl prodrug approach initially developed for BMS-488043 was applied to BMS-626529, leading to the BMS-663068 prodrug as a phosphonooxymethyl tris(hydroxymethyl)-aminomethane salt. The conversion of BMS-663068 prodrug (fostemsavir) to the parent drug BMS-626529 was efficient after oral administration with virtually no prodrug detectable in the plasma. However, the half-life of BMS-626529, delivered as BMS-663068, was only 1.5 h, necessitating administering the drug at least every 8 h in order to achieve the target plasma concentration. Hence, an extended-release formulation was developed to overcome this obstacle (for details of the extended-release formulation development for BMS-663068, see reference [44]). 

The extended-release formulation of prodrug BMS-663068 was first evaluated in an 8-day monotherapy Phase 1b clinical trial where good overall efficacy was demonstrated [45]. Subjects with baseline susceptibility (IC50 < 100 nM toward the parent drug BMS-626529 by using the PhenoSense HIV-1 Entry Assay) responded well, leading to a maximum median decline between 1.21 and 1.73 log_10_ copies/mL. The successful proof-of-concept Phase 1b clinical trial enabled the initiation of a Phase 2b clinical trial where BMS-663068 in combination with TDF and RAL (see Table 1 for drug name abbreviation) was compared to ritonavir-boosted ATV in combination with TDF and RAL in treatment-experienced HIV-1 patients. At week 24, it was observed that the two arms of the Phase 2b study had comparable efficacy at achieving a viral load of <50 copies/mL, leading to a conclusion that BMS-663068 has the desired properties of safety and efficacy to proceed into late-stage clinical studies [46]. In a pivotal Phase 3 clinical trial (BRIGHTE; NCT02362503), highly treatment-experienced patients with multidrug resistance were recruited to test the efficacy of BMS-668063 [47]. Success was achieved at the primary endpoint where patients treated with BMS-663068 in combination with failing regimen exhibited statistical superiority in viral reduction over placebo added to failing regimen over an 8-day treatment window (0.79 log_10_ copies/mL versus 0.17 log_10_ copies/mL; *p* < 0.0001). 

## 4. Mode of Action and Structural Basis of Inhibition for Temsavir and Analogues

Early studies following the identification of inhibitors of the temsavir family demonstrated the target of these inhibitors to be the HIV-1 envelope protein (see above). However, because the HIV-1 Env can undergo a series of conformational changes during viral entry, it was not immediately clear what was the mode of action for the temsavir family of compounds to achieve inhibition. Detailed biochemical studies showed that temsavir and its analogs bind to gp120 and stabilize gp120 in a conformation that is incapable of binding to the CD4 receptor [33,48]. It has also been suggested that under certain conditions, CD4 can bind to temsavir bound to gp120 [49]. However, in one way or another, temsavir binding prevents the conformational changes required for the eventual exposure of gp41 for fusion. 

Based on biochemical data, SAR insights, and resistance mutations observed in vitro and in vivo, computational models of temsavir related inhibitors in complex with HIV-1 gp120 have been proposed to provide a structural basis of the inhibition [50,51]. In these models, the temsavir inhibitor was predicted to bind adjacent to the CD4 binding loop and β20-β21 hairpin (site 2 in Figure 3b), providing a plausible mechanism where temsavir and related compounds stabilize the gp120 by preventing the rearrangement of the β20-β21 hairpin from forming the bridging sheet which is critical for CD4 receptor binding.

Cocrystal structures of the HIV-1 Env trimer ectodomain gp140 in complex with BMS-378806 and BMS-626529 (Figure 3c,d) were later determined to provide high-resolution structural details of inhibitor binding. Utilizing a prefusion stabilized HIV-1 Env ectodomain construct gp140 SOSIP, which was liganded with two neutralizing antibody fragments 35O22 and PGT122, complex structures were determined to a resolution of ~3.0–3.5 Å [53]. The cocrystal structures are consistent with the previous prediction that the inhibitor binding site is close to the CD4 binding loop and β20-β21 hairpin; however, several unexpected structural features were uncovered in the crystal structures. First, the inhibitors were mostly covered by the β20-β21 hairpin leaving only small solvent-accessible surfaces on the inhibitors in the complex structures (site 1 in Figure 3b). This binding mode suggests that the inhibitors might dock into the binding pocket when the β20-β21 hairpin transiently opens to sample a conformation compatible with CD4 binding. Once an inhibitor enters the binding pocket, it forms interactions with the β20–β21 hairpin, preventing (or reducing the frequency of) β20–β21 hairpin conformational changes. Secondly, as a consequence of the observed binding mode, the inhibitors prevent the formation of the water channel observed in the CD4-bound gp120 structure. Concurrently, due to the presence of the inhibitor, the sidechain of W427 was pushed into a position that was occupied by the sidechain of F43 in the CD4-bound gp120 structure, eliminating a site that accommodates the F43 of CD4. Thirdly, the azaindole heterocycle points toward the alpha-helix 2, making the inhibitor pose significantly different from the previously proposed computational models. In this orientation, the azaindole NH forms a hydrogen bond with the side chain of D113 on the α1 helix. Overall, the cocrystal structures, consistent with previously generated biochemical data and resistance mutations observed in vitro and in vivo, provided mechanistic insights into the HIV-1 inhibition by temsavir and related compounds.

## 5. Next Generation Inhibitors: Potent Temsavir Analogues 

Temsavir (administered as prodrug fostemsavir) is the first-in-class attachment inhibitor, acting through an inhibition mechanism distinct from other FDA-approved antiretroviral drugs. More potent derivatives or analogs of temsavir will provide added benefits of broader coverage and prevention of emergent resistance mutations by increasing genetic barrier, as demonstrated by temsavir (BMS-626529) when compared to early-generation clinical candidates, such as BMS-488043. 

BMS-818251 is a temsavir analog with >10-fold improved in vitro antiviral potency compared to temsavir, as demonstrated in a pseudotyped virus panel composed of 208 clinical HIV-1 strains [52]. With a cyano alkene replacing an amide in the benzoyl functional group of temsavir, and a thiazole substituent replacing the triazole in the 6-azaindole core, a prominent feature of BMS-818251 is a long extension from the thiazole (Figure 2). In a cocrystal structure of BMS-818251 in complex with gp160 SOSIP, this long extension forms productive interactions with D113, R429, and Q432 of Env from the BG505 strain, while preserving other interactions observed in the temsavir cocrystal structure, providing a structural basis of improved potency (Figure 3e). Notably, temsavir is very ineffective in inhibiting clade AE viruses, while BMS-818251 showed at least a 40-fold improvement in inhibiting clade AE viruses in the pseudotyped virus neutralization assay. Two additional compounds designated 87 and 88 have been proposed to be able to engage and interact with D113 or K117 through a long extension from the 6-azaindole core based on computation modeling [32]. Interestingly, the functional groups in compounds 87 and 88 extending from the 6-azaindole core were rather rigid in contrast to the more flexible functional group in BMS-818251. Initially, antiviral assays showed that these two compounds had a 10-fold more potent antiviral activity than temsavir in a small panel of pseudotyped viruses composed of clade B, C, and D HIV-1 strains.

The *N*,*N*’-difunctionalized piperazine building block in the temsavir family of inhibitors has been used to construct a library of chemical probes to study the dynamic properties of the β20-β21 hairpin as a regulatory switch in controlling Env structural rearrangement during entry [54]. Compound 484 (Figure 2) in this library has been determined to possess low micromolar activity toward a panel of 14 HIV-1 strains of clades A, B, C and D. Interestingly, despite the moderate inhibition activity, a cocrystal structure of compound 484 in complex with BG505.SOSIP showed that the piperazine ring of 484 occupied the same site within the binding pocket (site 1 in Figure 3b) as the piperazine ring in the BMS-818251 cocrystal structure [52], suggesting that the piperazine ring is a common building block for inhibitors that bind to this pocket beneath the β20-β21 hairpin.

A separate binding pocket in close proximity to the temsavir binding pocket has been identified as the binding site for several entry inhibitors, including CD4 mimicking small molecule inhibitors (site 1 in Figure 3b). The cocrystal structures of NBD-556 and DMJ-II-121 in complex with gp120 showed that these inhibitors bound to a cavity in between the β20-β21 hairpin and the CD4 binding loop [55,56,57]. This cavity would accommodate the Phe43 residue of the CD4 receptor in the gp120-CD4 complex structure and thus provides evidence that inhibitors directly related to NBD-556 and DMJ-II-121 inhibit HIV-1 by directly competing with CD4 binding. However, in general, NBD-556 and DMJ-II-121 have micromolar range antiviral activity while temsavir has nanomolar range potency. Overall, these structures highlight that two distinct binding pockets close to the β20-β21 hairpin are present for attachment inhibitor binding and can serve as a basis for the development of next-generation attachment inhibitors. Attachment inhibitors that exploit the features of both binding pockets might have superior antiviral activity than current attachment inhibitors.

## 6. Conclusions

The newly FDA-approved fostemsavir represents the first-in-class drug in the attachment inhibitor class for HIV-1 treatment. Due to the relatively high dose and the requirement for an extended-release formulation, it poses significant challenges to develop fostemsavir as part of combination antiretroviral therapy for early line treatment. However, the novel mode of inhibition with no cross-resistance to currently available antiretroviral classes and favorable drug-drug interaction profile has made the approval of fostemsavir an important advancement to address the unmet need in highly treatment-experienced HIV-1 patients who have limited treatment options due to multidrug resistance. Mechanistic studies elucidated that temsavir binds to a pocket close to the CD4 binding site, providing a structural basis of how inhibition of attachment is achieved. Further development of inhibitors in the temsavir family will need to address the low antiviral potency against clade AE viruses. A broad spectrum temsavir derivative or analog with improved overall antiviral potency will further increase the genetic barrier for emerging resistance mutations, leading to superior next-generation attachment inhibitors.

In this review, we focused the discussion on small molecule inhibitors that interfere with the CD4 attachment step; however, protein-based inhibitors are also being actively developed to achieve the same outcome—namely, the inhibition of HIV-1 entry by interfering with the interaction between HIV-1 Env and the CD4 receptor. CD4-binding site broadly neutralizing antibodies, such as VRC01 [58,59,60], N6 [61] (ClinicalTrials.gov Identifier: NCT03538626), and 3BNC117 [62,63,64], are currently at different stages of clinical trials for HIV-1 infection treatment and/or prevention. In addition, it has been shown that soluble CD4 ectodomains can prevent the interaction between HIV-1 Env and the genuine CD4 receptors located on the target cell surface. An engineered fusion protein composed of human immunoglobulin constant regions and CD4 domains 1 and 2, designated CD4-Ig (and the improved version eCD4-Ig), have shown extraordinary breadth and potency against HIV-1 isolates [65,66,67]. A related strategy of utilizing CD4 decoys displayed on nanoparticle surfaces has been recently reported [68]. Protein-based attachment inhibitors, together with the small-molecule attachment inhibitors reviewed here, illustrate the attachment step of HIV-1 entry to be a valuable target in developing an armory to address the ever-increasing need against HIV-1 infection. 

## Figures and Tables

**Figure 1 viruses-13-00843-f001:**
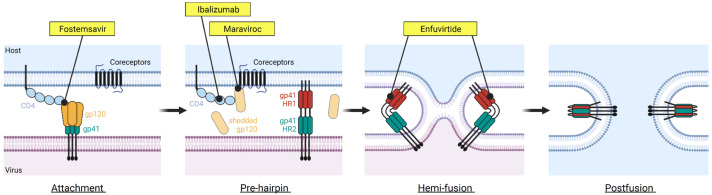
The HIV-1 entry process and FDA-approved drugs inhibiting this process. The HIV-1 entry is a complex process involving many steps. Four FDA-approved drugs targeting different steps of the HIV-1 entry are shown. These include fostemsavir that blocks CD4 binding (attachment inhibitor class), ibalizumab targeting the domain II of CD4 receptor (post-attachment inhibitor class), maraviroc binding to CCR5 coreceptor (CCR5 antagonist class), and enfuvirtide binding to gp41 (fusion inhibitor class). Figure was prepared in BioRender.

**Figure 2 viruses-13-00843-f002:**
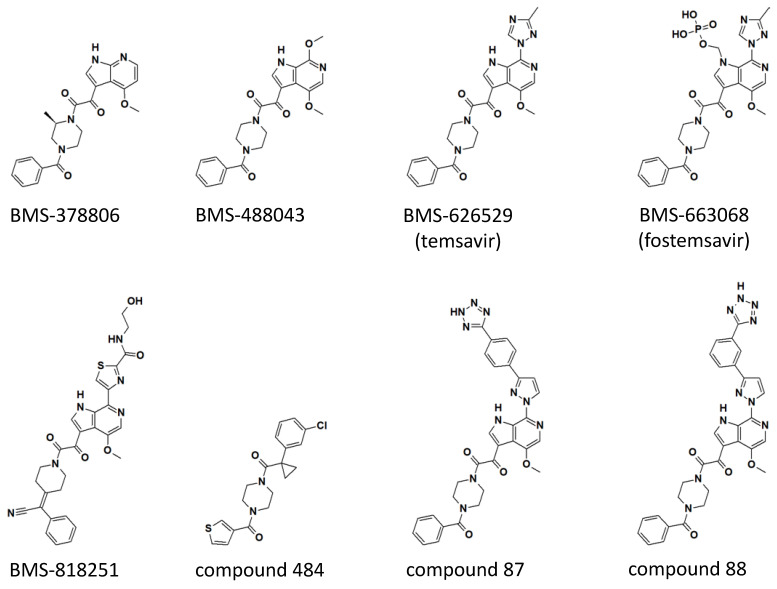
Chemical structures of temsavir and analogues. Two-dimensional chemical structures are shown for each compound. Three-dimensional structures of BMS-378806, BMS-626529 (temsavir), BMS-818251, and compound 484 in complex with HIV-1 envelope protein have been determined (see Figure 3). Compounds 87 and 88 were based on the numbering from reference [32]. Computational models of compounds 87 and 88 in complex with HIV-1 Env are also available in reference [32].

**Figure 3 viruses-13-00843-f003:**
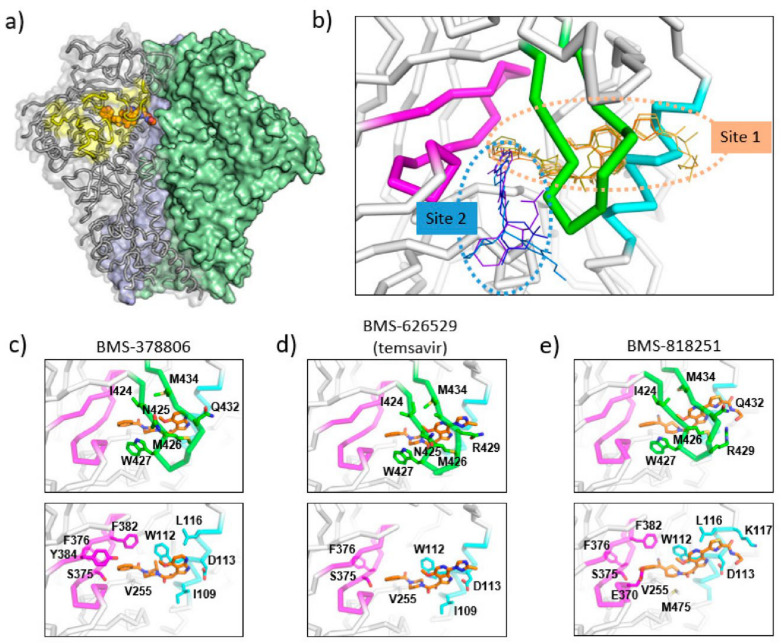
Binding sites of temsavir and related compounds on HIV-1 Env. (**a**) One protomer of the Env trimer is shown as ribbon with transparent surface, while the other two protomers are shown as solid surfaces in green and blue. CD4 binding site is shown as yellow surface patch. A temsavir analog BMS-818251 is represented as orange spheres. (**b**) Two partially overlapping but distinct inhibitor binding sites are present in the vicinity of CD4 binding site. Inhibitors with available complex structures are shown superimposed. Inhibitors binding to site 1 are colored in orange/brown shades and include BMS-626529 (temsavir; PDB: 5U7O), BMS-378806 (PDB:6MTJ), BMS-818251 (PDB:6MU7), BMS-814508 (PDB:6MU6), BMS-386150 (PDB:6MU8) and compound 484 (PDB:6MTN). Inhibitors binding to site 2 are colored in blue/purple shades and include NBD-556 (PDB:3TGS), NBD-10007 (PDB: 4DKV), and DMJ-II-121 (PDB: 4I53). Detailed interactions between HIV-1 Env and BMS-378806, BMS-626529 (temsavir), and BMS-818251 are shown in panels (**c**–**e**), respectively. b20-b21 hairpin (residues 423–436) is shown as green ribbon. The C-terminus of a1 helix (residues 107–117) is shown as cyan ribbon, and part of CD4-binding loop (residues 369–385) is shown as magenta ribbon. The b20-b21 hairpin is removed in the lower panels of (**c**–**e**) for clarity. HIV-1 Env residues that interact directly with inhibitors are shown as sticks with residue type and number labeled. Figure 3e was originally published in *Nat Commun* 10, 47 (2019) [52].

**Table 1 viruses-13-00843-t001:** Currently approved antiretroviral drugs for HIV-1 treatment.

Drug Class		Generic Name (Other Names and Acronyms)	Brand Name
NRTIs		abacavir (abacavir sulfate, ABC)	Ziagen
		emtricitabine (FTC)	Emtriva
		lamivudine (3TC)	Epivir
		tenofovir disoproxil fumarate (tenofovir DF, TDF)	Viread
		zidovudine (azidothymidine, AZT, ZDV)	Retrovir
NNRTIs		doravirine (DOR)	Pifeltro
		efavirenz (EFV)	Sustiva
		etravirine (ETR)	Intelence
		nevirapine (extended release nevirapine, NVP)	Viramune/XR
		rilpivirine (rilpivirine hydrochloride, RPV)	Edurant
Protease Inhibitors		atazanavir (atazanavir sulfate, ATV)	Reyataz
		darunavir (darunavir ethanolate, DRV)	Prezista
		fosamprenavir (fosamprenavir calcium, FOS-APV, FPV)	Lexiva
		ritonavir (RTV)	Norvir
		saquinavir (saquinavir mesylate, SQV)	Invirase
		tipranavir (TPV)	Aptivus
Integrase Inhibitors		cabotegravir (cabotegravir sodium, CAB)	Vocabria
		dolutegravir (dolutegravir sodium, DTG)	Tivicay
		raltegravir (raltegravir potassium, RAL)	Isentress
Entry Inhibitors	Fusion Inhibitors	enfuvirtide (T-20)	Fuzeon
	CCR5 Antagonists	maraviroc (MVC)	Selzentry
	Post-attachment Inhibitors	ibalizumab-uiyk (Hu5A8, IBA, Ibalizumab, TMB-355, TNX-355)	Trogarzo
	Attachment Inhibitors	fostemsavir (fostemsavir tromethamine, FTR)	Rukobia

## Data Availability

No new data were created or analyzed in this study. Data sharing is not applicable to this article.

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
