# Peer review of "Small Molecule HIV-1 Attachment Inhibitors: Discovery, Mode of Action and Structural Basis of Inhibition"

_viruses, 2021, doi:10.3390/v13050843_

Round 1
Reviewer 1 Report
This is a very nice review on the Fostemsavir, a first-in-class HIV-1 attachment inhibitor approved by the FDA in July 2020. It could be published as it is.
Author Response
I thank the reviewer's very positive feedback.
Reviewer 2 Report
This was an informative article about HIV-1 attachment inhibitors. The authors did a good job going through the details of and comparing a number of newly developed compounds that block HIV-1 attachment. I found the emphasis on chemistry and chemical derivatives of the compound a little distracting at times but I will leave it to the authors to decide whether to change or not.
Minor typos:
Line 104: gloxamide-->glyoxamid
Line 130: compared the original --> compared to the original
Line 266: a inhibitor--> an inhibitor
Author Response
I appreciate the careful reading of the manuscript by the reviewer. The typos in the specific lines pointed our by the reviewer are corrected. To maintain the high level of accuracy that is consistent with the Viruses journal style, I will retain the complete description of chemistry regarding fostemsavir and derivatives in this manuscript.